

# A stable Faroe Bank Channel overflow 1995-2015

Bogi Hansen[1], Karin Margretha Húsgarð Larsen[1], Hjálmar Hátún[1], Svein Østerhus[2]

[1]Faroe Marine Research Institute, PO Box 3051, FO-110 Tórshavn, Faroe Islands
[2] Uni Research Climate, Nygårdsgata 112, N-5008 Bergen, Norway

5   *Correspondence to*: Bogi Hansen (bogihan@hav.fo)

**Abstract.** The Faroe Bank Channel is the deepest passage across the Greenland-Scotland Ridge (GSR), and through it, there is a continuous deep flow of cold and dense water passing from the Arctic Mediterranean into the North Atlantic and further to the rest of the World oceans. This *FBC-overflow* is part of the Atlantic Meridional Overturning Circulation (AMOC), which has recently been suggested to have weakened. From November 1995 to May 2015, the FBC-overflow has been monitored by a continuous ADCP (Acoustic Doppler Current Profiler) mooring, which has been deployed in the middle of this narrow channel. Combined with regular hydrography cruises and several short-term mooring experiments, this allows us to construct time series of volume transport and to follow changes in the hydrographic properties and density of the FBC-overflow. The mean *kinematic overflow*, derived from the velocity field solely, was found to be (2.2±0.2) Sv (1 Sv = $10^6$ m$^3$ s$^{-1}$) with a slight, but not statistically significant, positive trend. The coldest part, and probably the bulk, of the FBC-overflow warmed by a bit more than 0.1 °C, especially after 2002. This warming was, however, accompanied by increasing salinities, which seem to have compensated for the temperature-induced density decrease. Thus, the FBC-overflow has remained stable in volume transport as well as density during the two decades from 1995 to 2015. This is consistent with reported observations from the other main overflow branch, the Denmark Strait overflow, and the three Atlantic inflow branches to the Arctic Mediterranean that feed the overflows. If the AMOC has weakened during the last two decades, it is not likely to have been due to its northernmost extension – the exchanges across the Greenland-Scotland Ridge.

Keywords: Overflow, AMOC, in situ observations

## 1 Introduction

25   Overflow is the term generally used to describe the bottom-intensified flow of cold and dense water from the Nordic Seas through the deep passages across the Greenland-Scotland Ridge (GSR) to the North Atlantic (Saunders, 2001). The strongest overflow branch in terms of volume transport passes through the Denmark Strait. This *DS-overflow* is estimated to transport around 3.5 Sv (1 Sv = $10^6$ m$^3$ s$^{-1}$) of dense ($\sigma_\theta > 27.8$ kg m$^{-3}$) water (Jochumsen et al., 2012; Harden et al., 2016). A similar amount of overflow is generally considered to pass east of Iceland in three branches where the overflow across the Wyville-





Thomson Ridge is weak (< 0.3 Sv) (Østerhus et al., 2008). The overflow across the Iceland-Faroe Ridge (IFR-overflow) is not well constrained by observations despite considerable effort (Beaird et al., 2013; Olsen et al., 2016).

The third overflow branch east of Iceland is the deep flow through the Faroe Bank Channel (FBC), FBC-overflow, which is the main focus of this study. With a sill depth of 840 m, the FBC is by far the deepest passage across the GSR and one of the main pathways for the cold and dense overflow waters generated in the Arctic Mediterranean (Fig. 1). With an average volume transport close to 2 Sv, the FBC-overflow is generally estimated to contribute about one third of the total overflow, and is second only to the DS-overflow (Østerhus et al., 2008).

The FBC-overflow includes the densest water to pass the GSR, although entrainment and mixing with other water masses after passing the ridge makes the end product less dense than the Denmark Strait overflow. If there is no significant detrainment (Mauritzen et al., 2005), the entrainment of ambient waters should increase the volume transport of the FBC-overflow on its passage from the FBC to the Deep Western Boundary Current in the Labrador Sea. This would make it one of the main contributors to the formation of NADW and the lower limb of the Atlantic Meridional Overturning Circulation (AMOC). The FBC-overflow transports (oxygen and) anthropogenic carbon dioxide absorbed from the atmosphere into deeper parts of the World Ocean (Sabine et al., 2004) and it is one of the processes that generates the driving force for the warm Atlantic inflow to the Arctic Mediterranean (Hansen et al., 2010).

The FBC-overflow is therefore an important component of the ocean circulation and climate system and it has been the focus of many different studies through the years (e.g., Hermann, 1967; Borenäs and Lundberg, 1988; Saunders, 1990; Hansen and Østerhus, 2007; Olsen et al., 2008). Many attempts have also been made to determine its volume transport as reviewed by Saunders (2001), Hansen and Østerhus (2000), and Hansen and Østerhus (2007). Most of them are consistent with the value presented by Hansen and Østerhus (2007). For the 1995 to 2005 period, they estimated the average volume transport of kinematic overflow, derived solely from the velocity field, to be (2.1±0.2) Sv whereas the volume transport of dense ($\sigma_\theta > 27.8$ kg m$^{-3}$) FBC-overflow was (1.9±0.3) Sv. We will frequently cite results from this study and will hereafter refer to it as: "HØ2007".

In the present study, we do not aim towards refining this estimate. Rather, we will study the variations, mainly to see whether there are any systematic changes. The main motivation for this is the link between the FBC-overflow and the AMOC. Climate models have long projected AMOC weakening (Collins et al., 2013) and recently there have been reports that this weakening has already started (Smeed et al., 2014; Robson et al., 2014; Rahmstorf et al., 2015).

A weakened AMOC could be due to weaker deepwater formation in the western North Atlantic, especially the Labrador Sea (Robson et al., 2014), but Smeed et al. (2014) found that their observed 2004-2012 decrease was due to: "a decrease in the southward flow of lower North Atlantic deep water below 3000 m", which they consider to be fed from the overflows.

Motivated by this, the main aim of this study is to analyze our observational data set from the FBC, initiated in November 1995, to see whether there are any indications of a weakening or other systematic changes. The observations



include continuous monitoring of the velocity field with moored ADCPs (Acoustic Doppler Current Profilers), regular hydrography cruises, and short-term dedicated mooring experiments.

Following HØ2007, our main time series for overflow transport is the kinematic overflow. This parameter is based on the velocity field, solely, and can be derived from the ADCP measurements. This parameter does not, however, provide a complete picture of the overflow since its temperature and salinity characteristics could vary even if the kinematic overflow does not and especially the density of the overflow may affect the AMOC.

A number of studies have shown that the density-driven circulation in Stommel's (1961) simple model does not provide a complete explanation of AMOC forcing (e.g., Munk and Wunsch, 1998; Toggweiler and Samuels, 1998). Nevertheless, model studies (Griesel and Maqueda, 2006; Roberts et al., 2013) have linked the meridional density gradient and its associated pressure gradient at depth with AMOC intensity. In addition to volume transport of the FBC-overflow, we therefore use our observations to investigate the variations and changes in the hydrography and especially the density of the FBC-overflow.

On its way from the FBC to the Deep Western Boundary Current, the FBC-overflow mixes with and entrains water from the Atlantic side of the GSR. The associated change in hydrographic properties has been well documented (e.g., Fogelqvist et al., 2003), but the change in volume transport is more difficult to estimate, which means that the contribution of modified FBC-overflow to NADW and to the AMOC is less well known. Here, we do not aim to solve that question, but we present some new observations and discuss the state of our understanding of this topic.

## 2 Material and methods

We use observations from moored instrumentation and from ship-borne CTD observations. The moorings have all been deployed over the sill (Fig. 2a). The CTD data are mainly from selected stations on three regularly occupied standard sections as well as an area in the FSC, all of them shown in Fig. 1. In this section, we also describe the method used for estimating kinematic overflow and some statistical methods.

### 2.1 Data from moored instrumentation

ADCP moorings have been deployed at four different sites along a section over the sill of the channel (Fig. 2). The dominant location is site FB, which has been occupied since November 1995 except for annual (2-3 weeks) servicing intervals and gaps due to mooring or instrument failure. At site FC, regular deployments were initiated in summer 2002 and have continued in most years since then. The other two sites have only had one deployment each, lasting 70 days at FA, and 364 days at FG. At most sites, the ADCP was located in the top of a traditional (very short) mooring, but the ADCP at site FG was located in a bottom mounted frame to protect it from fishing gear. A complete list of ADCP deployments is in a technical report (Hansen et al., 2015a), available online, which also lists details and discusses the quality of these measurements.



In addition to velocity profile, the ADCP measures temperature at the instrument. This sensor may have an offset of several tenths of a degree, but comparison to simultaneous CTD profiles allows a calibrated temperature series to be generated with accuracy of about 0.05 °C (HØ2007). After the summer of 2001, a SeaBird MicroCat (SBE37) instrument, which measures temperature more accurately, has been attached to the ADCP. The MicroCats have been regularly calibrated either at the factory or by being attached to a SeaBird 911plus CTD that is lowered and kept for some time at a depth with stable temperature. None of the calibrations have shown larger drift than a few millidegrees. Since July 2001, the bottom temperature series at FB therefore should be accurate within 0.01 °C, at least.

The MicroCat also measures conductivity, from which salinity may be derived, but evaluation of the data has generally concluded that the uncertainty of these measurements is too high to yield useful results, perhaps due to contamination in this bottom-near high-turbulence regime. Thus, we do not use these data here.

## 2.2 Data from CTD profiles

We mainly use CTD data from the period 1996 to 2015. These have been acquired with a SeaBird SBE911plus system with double temperature and conductivity sensors. Temperature has been calibrated at the factory annually and salinity has been calibrated ashore by salinometer (Autosal) analysis of water samples acquired in triples, generally at every profile. For the deep, weakly stratified, waters mainly considered here, the typical accuracy is estimated at 0.001 °C for temperature and 0.002 for salinity. These values may not hold for individual profiles, but for the averages, considered in this study, they should be representative. All salinity values are presented as practical salinities.

In order to relate the deepwater changes to changes in the upper layers, we also use time series of temperature and salinity in the cores of Atlantic water (defined by maximum salinity) on sections V and N, derived from de-seasoned CTD observations. These series are updated from Larsen et al. (2012).

## 2.3 Method for computing overflow volume transport

For the whole period considered here (November 1995 to May 2015), only one ADCP mooring site (FB, Fig. 2) has usually been occupied. Using the complete data set up to summer 2005, HØ2007 showed, however, that this one ADCP was sufficient to generate a time series of kinematic overflow, especially on time scales of a month or longer. This was to be expected since the deep parts of the FBC are narrow over the sill, with a width on the order of the baroclinic Rossby radius. The cross-channel variations of the along-channel velocity are highly correlated on time scales of months or longer (HØ2007).

By definition, kinematic overflow is the volume transport below the *interface*, which is defined from the velocity field (Supplementary Fig. S1), integrated across the channel. Based on the horizontal co-variation, HØ2007 developed algorithms to calculate kinematic overflow from the daily averaged velocity profile at site FB. This involved both inter- and extrapolation and especially the horizontal extrapolation towards the Faroe Plateau introduced some uncertainty.





With the new ADCP measurements at site FG (Fig. 2), this uncertainty can be reduced and the algorithms adapted. This has been done in the previously cited technical report (Hansen et al., 2015a) and a new time series of kinematic overflow has been generated for the whole period. For monthly averages, the correlation coefficient between the new and the old series is 0.92 and the overall averages only differed by 1 %. HØ2007 estimated the uncertainty of the kinematic overflow to be 0.2 Sv. With the adapted algorithm, this uncertainty has not increased and we will retain this value.

One problem with the method for determining kinematic overflow is that a barotropic current throughout the water column could introduce a bias in the estimate. To check this, we looked at the along-channel velocity at a fixed depth, above the overflow layer at site FB. Averaged over entire deployment periods for deployments lasting at least 11 months, this velocity varied between 6 cm s$^{-1}$ in the overflow direction (northwest) and 5 cm s$^{-1}$ in the opposite direction with no significant trend over the observational period (Fig. S2). This problem, therefore, will not have any effects on long-term trends in the estimated overflow transport.

## 2.4 Statistical methods

Several of the time series considered in this study may be seen as super-positions of slowly varying signals + seasonal signals + random variations. We use an *iterative decomposition method* to separate seasonal and long-term variations. The seasonal variation generally has a roughly sinusoidal shape and a simple analysis may be made by regressing the time series on a sinusoidal seasonal variation, where the phase lag is varied to give maximum correlation. The long-term variation may then be calculated as a running mean of de-seasoned values. From the determined long-term variation, a new estimate of seasonal variation can be achieved. This procedure is repeated iteratively and rapidly converges so that we get a seasonal signal that is not so much contaminated by long-term variations and we get a time series of 3-year running mean, which is the average of all the de-seasoned values within each 3-year period. This also allows us to calculate the standard error of each 3-year mean value.

We will investigate temporal trends for several different parameters by *linear trend analysis*, which is done by standard linear regression analysis of the parameter on time. In general, we will report the trend as the regression slope ± its 95 % confidence interval as determined by a standard t-test.

The statistical significance of standard errors in the iterative decomposition method and confidence limits in the trend analysis depend on assumptions of serial correlation and normal distributions, which may not always be valid. Therefore, we do not claim specific statistical confidence levels for the reported values, although in some cases we use them to claim lack of significance.



## 3 Results

### 3.1 Overflow volume transport

Monthly averaged kinematic overflow values (Fig. S3) ranged from 1.2 Sv to 3.2 Sv. In Fig. 3, the variations are split into 3-year running mean values (Fig. 3b) and deviations (anomalies) from these (Fig. 3a) as described in Sect. 2.4. As noted by

HØ2007, the transport anomalies have a seasonal variation with maximum in August. Also shown is the annually averaged transport excluding days number 136-195, during which period the servicing gap in different years has occurred. The overall average transport was 2.2 Sv and a linear trend analysis of this series gave (0.010±0.013) Sv year$^{-1}$. The annual averages (dashed curve in Fig. 3b) do not indicate strong serial correlation, but taking that into account can only increase the uncertainty. Thus, we see no statistically significant trend in the kinematic overflow and the indication is of a strengthening

rather than a weakening.

### 3.2 Bottom temperature

The main data set for overflow temperature is the time series of temperature measured by the ADCP until summer 2001 and by the attached Microcat after that. The instrument depth has varied slightly from one deployment to another, but has been close to a typical depth of 810 m. This depth is around 30 m shallower than the sill depth and site FB is also displaced some

3 km northeast of the deepest part of the sill (Fig. 2). The question therefore arises, how well this time series represents the bottom water over the sill.

   The effect of vertical displacement may be estimated by considering how the temperature varies with depth from 840 m upwards at standard station V06 (Fig. S4a). Station V06 is around 25 km upstream of the sill (Fig. 2a) and we may expect some change to the vertical structure as the overflow water accelerates towards the sill. Nevertheless, the bias introduced by

the vertical displacement is not likely to exceed a few hundredths of a degree. Also, the relatively small standard error (Fig. S4a) indicates that the bias should be constant within ±0.01 °C as long as we are considering temporally averaged values.

   Similarly, the effect of the horizontal displacement may be estimated by comparing the bottom temperature at FB with simultaneously (same day) measured temperature at 810 m depth at station V06 (Fig. S4b). The correlation coefficient was 0.69 with a regression coefficient of 0.69±0.19. This indicates that the bottom temperature at FB varies a bit less than

temperature at the same depth at V06. On average, the water at 810 m depth at V06 was (0.045±0.011) °C colder than the bottom water at FB.

   Altogether, we may conclude that the coldest water flowing over the sill is probably somewhat colder than the bottom temperature measured at FB and this bias may well be on the order of 0.05 °C. When averaged over long periods, especially over a year, this bias seems, however, to be fairly constant with an uncertainty on the order of 0.01 °C. Thus, variations

observed at FB ought to represent the bottom water within this uncertainty. In the following, we will use the measurements at FB as representing the bottom temperature at the FBC sill with this caveat.



The bottom temperature measurements at site FB were averaged to monthly values, excluding months with less than 28 days. With the method in Sect. 2.4, they were split into 3-year running mean bottom temperature (Fig. 4b) and the monthly deviations (anomalies) from these (Fig. 4a). The monthly anomalies indicate a seasonal variation with minimum temperature in August as previously noted (HØ2007). The 3-year running mean (Fig. 4b) has a clear positive trend, especially after 2002. The shaded area on Fig. 4b indicates the uncertainty of the 3-year running mean, estimated as ± one standard error over each 3-year period, but not smaller than the instrumental uncertainty, which is ±0.05 °C prior to summer 2001 and ±0.01 °C after that.

An objective estimate of the bottom temperature change at FB can be obtained by a linear trend analysis (regressing annually averaged bottom temperature on time). For the whole period 1996 to 2014, this gives a warming of (0.10±0.06) °C. Using only the measurements after Microcats were introduced in 2001, gives almost identical results. The high uncertainty before 2001 makes it difficult to ascertain the temperature variation in the early years, but Fig. 4b does not indicate that it was appreciably colder in this period than in 2002. The bottom temperature at FB, thus, most likely increased by a bit more than 0.1 °C during our observational period.

In the following, we will mainly use potential temperature to take into account depth changes. The potential temperature, θ, at the bottom at site FB is approximately equal to the in situ temperature minus 0.033 °C and will have increased by the same amount as the in situ temperature.

### 3.3 Salinity and density changes

By itself, a warming of the bottom water would imply reduced density, but density also depends on the salinity. Thus, we need to consider possible salinity changes of the FBC-overflow. Unfortunately, our salinity measurements over the sill of the FBC are not adequate to clarify this, but we have regular CTD measurements from regions close to and farther upstream of the sill. The FBC-overflow is fed from the deep layers of the FSC, but it experiences intensive mixing on its way towards and over the FBC-sill (Saunders, 1990; Mauritzen et al., 2005). Standard section V (Fig. 1) has only two stations that reach sufficient depths to cover the overflow (Fig. 2a) and their θ-S relationships are illustrated on Fig. 5 together with θ-S traces from two stations in the FSC.

The two stations from the FSC in Fig. 5 are on opposite sides of the channel (Fig. 1), but the data are only from cruises with occupations of both stations. The fact that their θ-S traces are almost indistinguishable (Fig. 5b) therefore shows that they represent the average conditions of the FSC waters feeding the FBC-overflow through this period (station S09 has not been regularly occupied since 2010). The two θ-S traces from section V are averages for the same period, although more frequently occupied. Their differences from the FSC θ-S traces therefore indicate substantial water mass changes occurring during the flow between the two sections.

Figure 5 also shows that the two θ-S traces from section V are different. Here, again, we only have used data where both stations were occupied during the same cruise (i.e., within a few hours) and so the lack of overlap between the shaded areas surrounding the traces in Fig. 5b implies that the difference is real. This difference has been ascribed to different





occurrence of a ''third water mass'' (Borenäs et al., 2001; Borenäs and Lundberg, 2004). However, both V05 and V06 show different θ-S relationships from the source waters in the FSC and HØ2007 argued that this was more likely due to more intensive mixing of the water arriving at V06.

The changes in θ-S relationship from the FSC to the FBC may involve both internal mixing within the overflow layer and admixing of upper layer waters. Temporal salinity changes in the FBC could therefore derive from changes in the upstream overflow waters or from local mixing of changing upper layer waters. In the following, we first consider the conditions on section V, slightly upstream of the sill, and then in the FSC and farther upstream. After that, we consider the effects of local mixing.

### 3.3.1 Salinity and density changes on section V

For the period 1996 to 2015, there are almost a hundred occupations of V05 and V06 and Table 1 lists overall changes of temperature, salinity, and potential density in different depth layers at these two stations during this period. The table shows increased temperatures and salinities at almost all depths, although some of the changes are not significantly different from zero.

For potential density, the changes in Table 1 are generally not significantly different from zero, but there are consistent tendencies. In the upper layers, the tendency is for density decrease. By the criterion $\sigma_\theta > 27.8$ kg m$^{-3}$, overflow water is generally found below 500 m at V05 and below 550 m at V06 (Table 1). In these layers, the tendencies are for density decrease at V05 and density increase at V06 although none of the changes are statistically significant.

A less noisy signal may be obtained by considering salinity changes at fixed potential temperatures (θ). In Fig. 6, this is done for three different values of θ within the overflow layer of section V. The time series are fairly noisy, but all of them indicate increasing salinity trends, which are confirmed by linear trend analysis (Table 2). The last three columns in Table 2 also show that to compensate for the salinity increases in the table, substantial increases in potential temperature are required, especially in the FSC.

### 3.3.2 Salinity and density changes upstream of the FBC

The FBC-overflow is fed from the deep waters of the FSC, which again are fed from the southern region of the Norwegian Sea. At fixed depth (e.g., 800 m) in these source waters, systematic changes in both potential temperature and salinity are seen, but no overall change in potential density (Fig. S5). These changes are part of a systematic change in the water mass structure of the FSC, in which the salinity minimum at intermediate temperatures has almost disappeared (Fig. 7a). For the period considered, the lowest salinities for fixed potential temperature were observed in 1997. After that, the salinity at all the potential temperatures in Fig. 7b started to rise. For high values of θ, the initial salinity increase was rapid, followed by almost stable conditions, whereas the colder waters exhibited a more gradual salinity increase, continuing throughout the period.



In the southern region of the Norwegian Sea, observations at 800 m depth on section N (Fig. 1) show water with similar potential temperature as the bottom water at FB and the two curves show a remarkable similarity in warming (Fig. 8).

### 3.3.3 The effects of local mixing

In addition to warming and salinification of the source water in the Norwegian Sea, the FBC overflow water will receive
heat and salt from the Atlantic layer on top by entrainment and mixing. Variations in the properties of the Atlantic water core in the FBC (Fig. S7) will then also induce variations in the overflow water.

An estimate of this contribution may be achieved by considering the changes in salinity from section S to section V (Fig. 5). According to HØ2007, the water with potential density exceeding 27.8 kg m$^{-3}$ on section V had a transport-averaged potential temperature $\theta = 0.25$ °C and salinity 34.93 for the 1996 to 2005 period. For the FSC, the available velocity
measurements in the deep water do not allow us to derive transport-averaged properties for the water flowing from the FSC into the FBC, but we can make a rough estimate by comparing average salinities at $\theta = 0.25$ °C for this period. At S08, this value was 34.893 i.e., 0.037 less saline than over the FBC sill. If this salinity increase along the flow is obtained by admixing upper layer Atlantic water with $\theta = 8.84$ °C and $S = 35.31$ (Average properties of Atlantic core 1996-2005), then 9 % of Atlantic water is required. This would also require that the overflow water in the FSC had an average potential temperature
of -0.59 °C, which seems rather cold, but the bottom-near water at S08 was on average -0.78 °C and, according to Mauritzen et al. (2005), this water also feeds the FBC-overflow.

Thus, an admixture of somewhere between 5 % and 10 % Atlantic water seems realistic. This implies that any change in temperature or salinity of the Atlantic water should be rapidly transferred to the overflow, although reduced by a factor of 5 to 10. From 1995 to 2003, the Atlantic water core in the upper parts of the FBC warmed by about 0.8 °C (Fig. 8a) and
increased in salinity by about 0.1 (Fig. S7). This may be expected to have warmed the FBC-overflow by 0.04 °C to 0.08 °C and increased its salinity by 0.005 to 0.01. After 2010, both temperature and salinity of the Atlantic water core on section V have decreased (Fig. S7).

From Fig. 8a, the variations in bottom temperature at FB are more similar to the temperature at 800 m on section N than a scaled down version of the Atlantic core temperature in the upper parts of the FBC. This would seem to indicate that
changing upstream conditions may be more important than local mixing, but a few more years of measurements are probably necessary to disentangle the effects of local mixing from the variations in the upstream source waters.

More important is the question, how the local mixing may have changed the density during our observational period. This is exemplified by the red and the green curves in Fig. 8b. There, we have assumed that the average properties during the 1996 to 2005 period, as cited above, came about as a mixture of cold water from the FSC and Atlantic water in the fraction 5
30   % (green) or 10 % (red), respectively. Assuming further that the mixing fractions as well as the cold water properties in the FSC remained unchanged, the red and green curves in Fig. 8b show how the density of this mixture would vary in time.

As seen in Fig. 8b, the overall change in density of the overflow water through our observational period from local mixing was small, but positive. The figure also shows density changes at two fixed depths in the source waters on section N



and there, as well, the density increased, although weakly. Although both source water variations and local mixing may have induced a warming of the overflow water, neither of them seems to have reduced its density; rather the opposite.

### 3.4 Downstream modification of FBC-overflow

It is not the intention in this study to provide a complete picture of the modification of the overflow water after it has crossed the FBC sill, but we take the opportunity to present some new CTD measurements that may be relevant in discussing the contribution of FBC-overflow to NADW and AMOC (Sect. 4.3). These measurements were obtained by RV *Magnus Heinason* on 20-21 May 2016 at eight stations (M1 to M8) along the red line in Fig. 9a.

On this occasion, waters colder than 3 °C, indicative of modified overflow, were seen close to bottom on all the stations except the southernmost (Fig. 9b). In Fig. 9c, the temperature is plotted against the height above bottom as measured by the altimeter mounted on the CTD. For most of the stations, the overflow layer is seen to be less than 40 m thick and for M7 less than 20 m. These thin layers are seen to be capped by very sharp thermoclines.

### 4 Discussion

The main aim of this study has been to determine whether the FBC-overflow has experienced any systematic changes during the observational period from Nov 1995 to May 2015, especially to ask whether there has been any change that could help explain reports of a weakened AMOC (e.g., Smeed et al., 2014). We have split this question into two parts: 1) has the kinematic overflow changed? and 2) have the properties (temperature or salinity and especially the density) of the overflow changed? In the next two sections, we summarize our results in an attempt to answer those two questions. We then briefly try to summarize our present understanding of the modifications that the overflow experiences from the FBC sill on its way towards the AMOC, followed by a discussion of the wider implications of our results.

### 4.1 Observed changes of the kinematic overflow

Monthly averaged kinematic overflow was calculated for all months with observational coverage (200 out of a total of 233 months) between December 1995 and April 2015. The calculations used the method described in HØ2007, slightly adapted based on new observations as detailed in Hansen et al. (2015a). The resulting time series (Fig. S3 and Fig. 3) did not exhibit any obvious systematic changes except perhaps a weak increase and a linear trend analysis gave (0.010±0.013) Sv year$^{-1}$.

Thus, we see no significant trend in kinematic overflow. If there has been a systematic change, it is furthermore most likely a strengthening. A weakening of the kinematic overflow through the period is unlikely.

### 4.2 Observed changes of overflow properties

Although the kinematic overflow, thus, seems to have been stable throughout the period, we do see changes in both temperature and salinity. Firstly, the bottom temperature has increased. Our temperature measurements prior to summer



2001 are too uncertain to allow definite conclusions, but they do not indicate any significant warming in that period. After initiation of Microcat measurements, we have high-quality continuous temperature measurements close to the bottom at site FB. As argued in Sect. 3.2, the temperature at this site may be a few hundredths of a degree warmer than the very coldest bottom water over the sill, but with an almost constant bias.

The warming that we see in the bottom water at site FB after 2001 should therefore be representative for the bottom temperature over the sill, which should represent the very coldest overflow crossing the GSR. This warming seems to have started around 2003 (Fig. 4b) and from then on until the end of the period, we see a total warming of a bit more than 0.1 °C.

    The deepest part of the overflow layer is fairly homogeneous (Fig. S4a). For the 1995-2005 period, more than 60 % of the overflow ($\sigma_\theta > 27.8$ kg m$^{-3}$) had potential temperatures below 0 °C (Table 6 in HØ2007). It therefore seems likely that

the warming at the bottom is also representative for much of the deepest part of the overflow layer. It is not clear, to what extent this warming is representative for shallower levels in the overflow layer, but it seems likely that there has been a general warming, although our CTD data have low signal to noise ratios for temperature in the deep water (Table 1).

    A warming at constant salinity implies a density decrease, but even a small salinity increase may be sufficient to offset a warming at low temperatures. Thus, a warming from -0.5 °C to -0.4 °C would be more than compensated for by a salinity

increase from 34.900 to 34.906 and there are clear indications that the overflow water has experienced salinity increase at all levels. This is most clearly seen by considering salinity change at fixed potential temperatures on section V (Fig. 6) as well as in the FSC (Fig. 7), with the changes generally increasing as we go from colder to warmer water and as we go from section V to the FSC (Table 2).

    The salinity and temperature increases will have opposite effects on density and the last three columns in Table 2 list

the temperature increases that would be required to compensate for the observed salinity increases on section V and in the FSC. For the bottom water ($\theta = -0.4$ °C), the required temperature increase (average of V05 and V06 in Table 2) is very similar to the maximum warming at site FB (Fig. 4b). Thus, the deep parts of the overflow, which comprise more than 60 %, seem to have maintained an almost constant density. For the upper parts of the overflow, we do not have good information on the warming rate, but from Table 2, considerable warming would have been needed to compensate for the density

increases induced by the observed salinity increases.

    The bottom temperature over the FBC sill follows the temperature at 800 m on section N in the southern Norwegian Sea (Fig. 1) fairly well (Fig. 8a) and the salinity at 800 m depth on station S08 in the FSC also varies synchronously with the salinity at 800 m depth on section N (Fig. S6b). This indicates that the water around 800 m depth on section N may be an important upstream source for the FBC-overflow. From the rather short time series available, it looks as if the temperature

and salinity variation at 800 m depth on section N is a reduced (by a factor of 5-10) and lagged (5-10 years) response to the Atlantic inflow (Fig. S6).

    Longer time series are available at station (Ocean Weather Ship) M at 66° N and 2° E, but the properties (especially salinity) at 800 m or 1000 m depth at station M are not similar to those at section N or in the FSC (Fig. S8). The link between the properties of Atlantic inflow and FBC-overflow has been addressed by Eldevik et al. (2009) and will not be



discussed further here. We only note that a change in the properties of the upstream source waters (e.g. at 800 m depth on section N) is one obvious mechanism for changing the properties of the FBC-overflow.

Another mechanism acts through local mixing with the Atlantic water, which is above the overflow water. According to Mauritzen et al. (2005), much of this mixing occurs in the basin east of the Wyville-Thomson Ridge and a rough estimate

says that 2 Sv of overflow water should not require many months to pass through this volume. So this mechanism should be very rapid. Disentangling the relative importance of these two mechanisms is not obvious from the present data set. Regardless of, which mechanism dominates, it seems clear, however, that the density change induced by increased temperatures and salinities of the FBC-overflow has not been a reduction (Fig. 8b).

### 4.3 Overflow modification

On its way from the FBC to the Deep Western Boundary Current off North America, the overflow water mixes with ambient water masses so that the resulting modified overflow is warmed by several degrees. This water mass, termed Iceland Scotland Overflow Water (ISOW), is usually defined by its density: $\sigma_\theta > 27.8$ kg m$^{-3}$ (e.g., Dickson and Brown, 1994; Saunders, 1994, 1996; Fogelqvist et al., 2003; Kanzow and Zenk, 2014).

In the simplest scheme, the overflow warms from close to 0 °C at the sill of the FBC up to about 3 °C in the Labrador

Sea, which may be explained by entrainment of equal amounts of ambient water, originating from the Atlantic side of the GSR, with temperatures around 6 °C (Hansen et al., 2004). By multivariate analysis of hydrographic, nutrient and halocarbon tracer data collected in July–August 1994, Fogelqvist et al. (2003) concluded that the ISOW contained 46 % of entrained water already in the Iceland Basin. Since additional entrainment may be expected, it has generally been assumed that the ISOW is a fairly equal mixture of original overflow water and water entrained after passing the ridge.

If there is no detrainment of overflow water into the ambient waters, this implies a doubling of volume transport, so that FBC-overflow after modification should contribute approximately 4 Sv to the ISOW. This value will, of course, be reduced if there is substantial detrainment.

The water mass transformation is especially intensive in the region immediately downstream of the FBC where the cold overflow meets and entrains the much warmer ambient water from the Atlantic. From their detailed survey, Mauritzen et al.

(2005) found that "the entrainment is sufficient to cause an approximate doubling of the transport ..." within 100 km of the sill and they did not find any evidence of detrainment. Based on measurements from a series of moored arrays, Geyer et al. (2006) reported that the mixing processes involved highly periodic oscillations with periods of a few days, which have been further discussed in a number of studies (Darelius et al., 2011, 2013, 2015) and seem to be the manifestation of baroclinic instability (Guo et al., 2014; Darelius et al., 2015).

These studies have done much to clarify the processes that modify the FBC-overflow immediately downstream of the channel. Quantifying the effects of entrainment/detrainment on the transport of the modified overflow plume will, however, probably require long-term observations from moored arrays that can determine transport in various density (temperature) classes. Geyer et al. (2006) did not attempt this, and neither did Darelius et al. (2011), but Ullgren et al. (2016) have



presented such an attempt. Based on year-long measurements on two mooring arrays, they found that detrainment from the overflow plume downstream of the FBC sill was of comparable magnitude to the entrainment and they only measured (1.7±0.7) Sv of "modified overflow" (colder than 6 °C) through their westernmost array 85 km downstream of the sill.

This result indicates that the assumption of substantial transport increase due to entrainment may not be valid and it could even indicate transport decrease. That conclusion would, however, be premature since it appears that the westernmost mooring array in Ullgren et al. (2016) did not cover the entire plume of overflow water. This is demonstrated in Fig. 9, which is based on a CTD survey conducted in May 2016 to check this very question. Of the eight CTD stations occupied on this cruise, M1 to M4 were at the positions where the four moorings of Ullgren et al. (2016) on this section were located (Fig. 9a) and the temperature distribution close to bottom clearly shows overflow south of their mooring array (Fig. 9b).

In their study, Ullgren et al. (2016) acknowledged that their westernmost array might miss a part of the overflow plume, but they assumed that this would not comprise a substantial fraction. That may indeed be the case but that is difficult to ascertain with the available information. Certainly, the overflow layer (< 3 °C) was thin (< 40 m) between stations M4 and M7 on 20-21 May 2016 (Fig. 9c), but this is close to the region where Mauritzen et al. (2005) observed near-bottom velocities exceeding 1 m s$^{-1}$ at these depths.

If this layer moves downhill, the isotherms (and isopycnals) slope the wrong way for it to be in geostrophic balance. Such a non-geostrophic flow would be expected to have a fairly high cross-isobath velocity component and, likely, this water feeds the deeper branch suggested by Hansen and Østerhus (2000) and confirmed by Geyer et al. (2006) (moorings A2 and A3 in Fig. 9a) and Beaird et al. (2013). It is not known whether the two branches later converge or some of the non-geostrophic flow continues deepening. Certainly, de Boer et al. (1998) found evidence of modified overflow water all the way to the depths of the Iceland Basin, although in highly variable concentrations.

Ullgren et al. (2016) have raised the point that transport increase of FBC-overflow due to entrainment may be much smaller than previously assumed, but there are open questions that ought to be checked with additional mooring deployments. From Fig. 9c, such deployments will, however, need to have instrumentation that can provide detailed near-bottom velocity profiles.

The relative roles of entrainment and detrainment immediately downstream of the FBC, thus, seem unresolved, at present. Farther downstream, Saunders (1996) reported (3.2±0.5) Sv of modified overflow ($\sigma_\theta > 27.8$ kg m$^{-3}$) through his mooring array south of Iceland while Kanzow and Zenk (2014) found (3.8±0.5) Sv (0.4 Sv of which was re-circulated) through their array over the eastern slope of the Reykjanes Ridge. Both of these estimates also include some IFR-overflow, although some of that will not satisfy the $\sigma_\theta > 27.8$ kg m$^{-3}$ criterion. Thus, these studies indicate that the FBC contribution to ISOW is less than a doubling of the original 1.9 Sv of FBC-overflow denser than 27.8 kg m$^{-3}$ (HØ2007).

Less dense ($\sigma_\theta < 27.8$ kg m$^{-3}$) components of modified FBC-overflow may, however, also contribute to the AMOC, e.g. through entrainment into the DS-overflow. For the meridional overturning circulation across a section along 59.5° N (southern tip of Greenland), Sarafanov et al. (2012) found the boundary between the upper (northward flowing) and deeper (southward flowing) branches to be at $\sigma_\theta = 27.55$ kg m$^{-3}$. This could help explain how both Dickson and Brown (1994) and



Sarafanov et al. (2012) by quite different methods find a total southward transport of dense ($\sigma_\theta > 27.8$ kg m$^{-3}$) water out of the Iceland and Irminger basins to be more than 13 Sv.

There is also the question, whether the mooring arrays over the northern slope of the Iceland Basin (Saunders, 1996; Kanzow and Zenk, 2014) cover the entire ISOW transport. Certainly, CFC-11 inventories presented by Smethie and Fine (2001) indicated much higher ISOW production rates and an updated estimate by LeBel et al. (2008) includes 5.7 Sv of ISOW with 55 % being entrained water. This was for the 1970-1997 period, i.e. before our observations, but Olsen et al. (2008) do not find the Iceland-Scotland overflow to have weakened since 1970.

To some extent, this discrepancy may be caused by the highly variable overflow component at the very depths of the Iceland Basin (de Boer et al., 1998). According to van Aken (2000), ISOW in the Iceland Basin is found below the Lower Deep Water, which partly derives from Antarctic Bottom Water (AABW). This is supported by the bottom-near oxygen maximum shown by Sarafanov et al. (2012) (their Fig. 5c), but apparently, the mooring arrays (Saunders, 1996; Kanzow and Zenk, 2014) have not covered this flow adequately.

This is also linked to the further pathways of ISOW. Originally, it was thought that the ISOW had to flow through the Charlie-Gibbs Fracture Zone in order to pass from the Eastern to the Western Basin (Dickson and Brown, 1994), but Saunders (1994) only found (2.4±0.5) Sv of ISOW ($\sigma_\theta > 27.8$ kg m$^{-3}$) to follow this path. Based on float trajectories, Kanzow and Zenk (2014) have shown, however, that there are alternative paths.

In addition to this, part of the ISOW continues southwards in the Eastern Basin and may be traced all the way to the Madeira Abyssal Plain (van Aken, 2001). LeBel et al. (2008) have estimated that as much as one third of the ISOW may take this pathway.

Summarizing, the contribution of modified FBC-overflow and Iceland-Scotland overflow as a whole to NADW and AMOC seems still not to be well quantified. According to some estimates, entrainment of water from the Atlantic side of the GSR more than doubles the volume transport, whereas other studies indicate much smaller transports.

**4.4 FBC-overflow and AMOC**

We now return to the question raised in the introduction. How do our observations of a stable FBC-overflow since 1995 fit with the claim by Smeed et al. (2014) of a significant weakening from 2004 to 2012 of the transport through the RAPID array at 26° N of lower NADW (LNADW), which they consider to be fed by the overflows?

The FBC-overflow is, of course, just one overflow component, accounting for only about one third of the total overflow, but the other main component, the DS-overflow, has also been reported to have no significant trend in transport Jochumsen et al. (2012, 2015). A substantial reduction in total overflow would also require compensation in some other component of the Arctic Mediterranean water balance. The main inflow to this region is the Atlantic inflow to the Nordic Seas, which has three branches. All of these have been monitored since the mid-1990s and none of them shows any sign of weakening (Jónsson and Valdimarsson, 2012; Hansen et al., 2015b; Berx et al., 2013) while the inflow from the Pacific has increased from 2001 to 2011 (Woodgate, 2012). A weakening of the total overflow might conceivably be compensated by

increased outflow in some other branch, such as the low salinity flow over the East Greenland Shelf, which is badly constrained by observations. Such a hypothetical increase would, however, have to be relatively high since the total overflow is the main outflow from the Arctic Mediterranean.

Smeed et al. (2014) suggest a possible explanation involving increased storage of LNADW north of the RAPID array during the period of reduced LNADW flow associated with an uplift of isolines. Another possibility would be a strong reduction in the entrainment of waters from the Atlantic into the overflows, but that would be pure conjecture.

There is, however, also a more semantic twist to this problem. Smeed et al. (2014) define LNADW as water between 3000 m and 5000 m depth, presumably all across the Atlantic whereas division of the NADW into the contributions from various sources usually has been made in terms of water mass characteristics and density (Sect. 4.3). In a baroclinic ocean, isolines must necessarily slope and cross depth levels. To link the LNADW weakening through RAPID to weakened overflow contribution requires verification that the boundaries between the various components of NADW and between NADW and AABW across the Atlantic have not moved substantially during the RAPID period.

Whatever the reason for the discrepancy between the RAPID and the overflow measurements, our results clearly indicate that the FBC-overflow has remained stable in volume transport and in density since the mid-1990s. Thus our results are consistent with the general picture of stable exchanges across the GSR during the last two decades and no weakening of the northernmost extension of the AMOC.

**Data availability**

Most of the data used in this study are available online at www.envofar.fo

**Competing interests**

The authors declare that they have no conflict of interest.

*Acknowledgements.* The authors wish to thank captains and crew on the RV *Magnus Heinason* as well as Regin Kristiansen for unfailing support during measurements at sea and Ebba Mortensen for data processing. Funding for the in situ measurements has been obtained from the Environmental Research Programme of the Nordic Council of Ministers (NMR) 1993–1998, from national Nordic research councils, from the Danish DANCEA programme, and from the European Framework Programs, lately under grant agreement No. GA212643 (THOR) and under grant agreement No. 308299 (NACLIM). Analysis and preparation of this manuscript was mainly funded by the NACLIM project.



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



**Table 1.** Overall change from 1996 to 2015 according to linear trend analyses for various depth layers at stations V05 and V06. For each layer, the table shows average potential density (Avg.$\sigma_\theta$), number of CTD occupations (N), and changes in temperature ($\Delta T$), salinity ($\Delta S$), and potential density ($\Delta\sigma_\theta$).

| Depth layer (m-m) | Average density and changes at station V05 | | | | | Average density and changes at station V06 | | | | |
|---|---|---|---|---|---|---|---|---|---|---|
| | Avg.$\sigma_\theta$ (kgm$^{-3}$) | N | $\Delta T$ ($^\circ$C) | $\Delta S$ | $\Delta\sigma_\theta$ (kgm$^{-3}$) | Avg.$\sigma_\theta$ (kgm$^{-3}$) | N | $\Delta T$ ($^\circ$C) | $\Delta S$ | $\Delta\sigma_\theta$ (kgm$^{-3}$) |
| 100-199 | 27.386 | 97 | 0.6±0.4 | 0.067±0.029 | -0.041±0.053 | 27.372 | 95 | 0.7±0.3 | 0.079±0.027 | -0.047±0.048 |
| 200-299 | 27.438 | 97 | 0.6±0.4 | 0.071±0.028 | -0.040±0.048 | 27.417 | 96 | 0.7±0.3 | 0.081±0.026 | -0.042±0.038 |
| 300-399 | 27.516 | 97 | 0.6±0.7 | 0.070±0.040 | -0.037±0.068 | 27.455 | 96 | 0.7±0.5 | 0.079±0.031 | -0.045±0.053 |
| 400-499 | 27.721 | 97 | 1.0±1.1 | 0.089±0.055 | -0.047±0.080 | 27.548 | 96 | 0.7±1.0 | 0.076±0.050 | -0.032±0.089 |
| 500-549 | 27.914 | 97 | 0.9±1.0 | 0.068±0.041 | -0.023±0.051 | 27.732 | 96 | 0.4±1.7 | 0.054±0.077 | 0.001±0.126 |
| 550-599 | 27.981 | 97 | 0.8±0.7 | 0.048±0.025 | -0.017±0.033 | 27.891 | 96 | 0.2±1.6 | 0.031±0.069 | 0.019±0.104 |
| 600-649 | 28.020 | 97 | 0.5±0.5 | 0.024±0.014 | -0.007±0.021 | 28.001 | 96 | -0.1±0.9 | 0.008±0.035 | 0.019±0.048 |
| 650-699 | 28.038 | 95 | 0.2±0.3 | 0.013±0.007 | -0.001±0.012 | 28.044 | 95 | -0.1±0.4 | 0.003±0.011 | 0.011±0.015 |
| 700-749 | 28.046 | 78 | 0.1±0.2 | 0.009±0.005 | 0.004±0.009 | 28.054 | 96 | 0.0±0.1 | 0.005±0.004 | 0.004±0.005 |
| 750-799 | | | | | | 28.056 | 85 | 0.1±0.1 | 0.006±0.003 | 0.002±0.004 |
| 800-849 | | | | | | 28.057 | 54 | 0.1±0.1 | 0.008±0.004 | 0.003±0.006 |





**Table 2.** Salinity increase at five fixed potential temperatures (θ) from 1996 to 2015 based on linear trend analyses for two stations on section V and stations in area A in the FSC (Fig. 1). The last three columns show the change in potential temperature that would be required to compensate for the salinity change with respect to potential density.

| θ | Salinity change 1996 to 2015 | | | Compensating θ-change | | |
|---|---|---|---|---|---|---|
| | V05 | V06 | FSC | V05 | V06 | FSC |
| -0.4°C | 0.011±0.006 | 0.005±0.003 | 0.013±0.004 | 0.18°C | 0.08°C | 0.21°C |
| +0.5°C | 0.021±0.007 | 0.021±0.015 | 0.032±0.006 | 0.27°C | 0.27°C | 0.41°C |
| +1.5°C | 0.030±0.013 | 0.033±0.018 | 0.050±0.014 | 0.32°C | 0.35°C | 0.52°C |
| +2.5°C | 0.030±0.018 | 0.036±0.022 | 0.049±0.022 | 0.27°C | 0.33°C | 0.44°C |
| +3.5°C | 0.024±0.023 | 0.033±0.018 | 0.044±0.027 | 0.19°C | 0.26°C | 0.35°C |



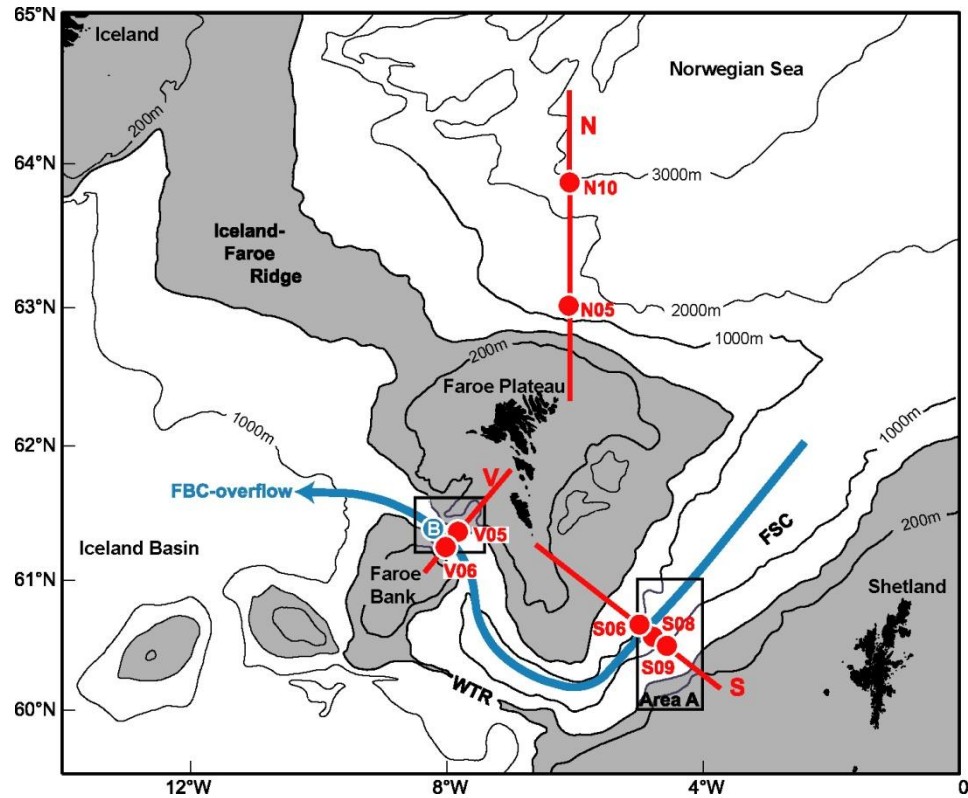

**Figure 1.** Map of the region with gray areas shallower than 500 m. Blue arrow indicates the path of the FBC-overflow through the Faroe-Shetland Channel (FSC), after which it is turned north-westwards by the Wyville-Thomson Ridge (WTR) to flow through the FBC between the Faroe Plateau and Faroe Bank. Red lines indicate three standard sections (V, S, and N) with selected standard CTD stations indicated by red circles. Blue circle labelled "B" indicates the long-term mooring site FB over the sill of the FBC. Black rectangle over the sill and section V shows area that is illustrated in more detail in Fig. 2. Black rectangle over section S shows area A, from which CTD stations deeper than 600 m have been analyzed.



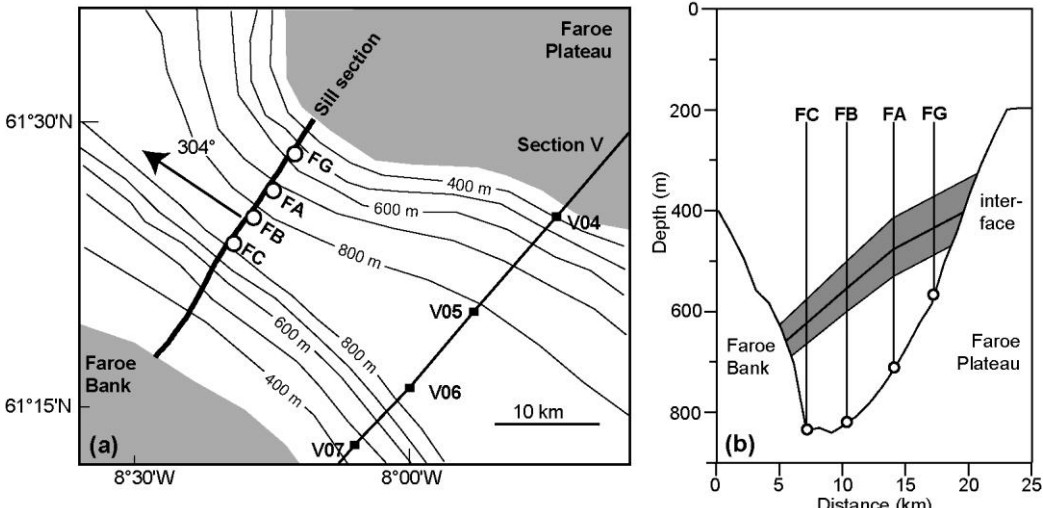

**Figure 2. (a)** Map showing the sill section with ADCP sites indicated by circles and the standard hydrographic section (section V) with standard stations indicated by black squares. The shaded area is shallower than 300 m **(b)** The sill section with ADCP sites indicated. The shaded area indicates a typical variation of the interface.





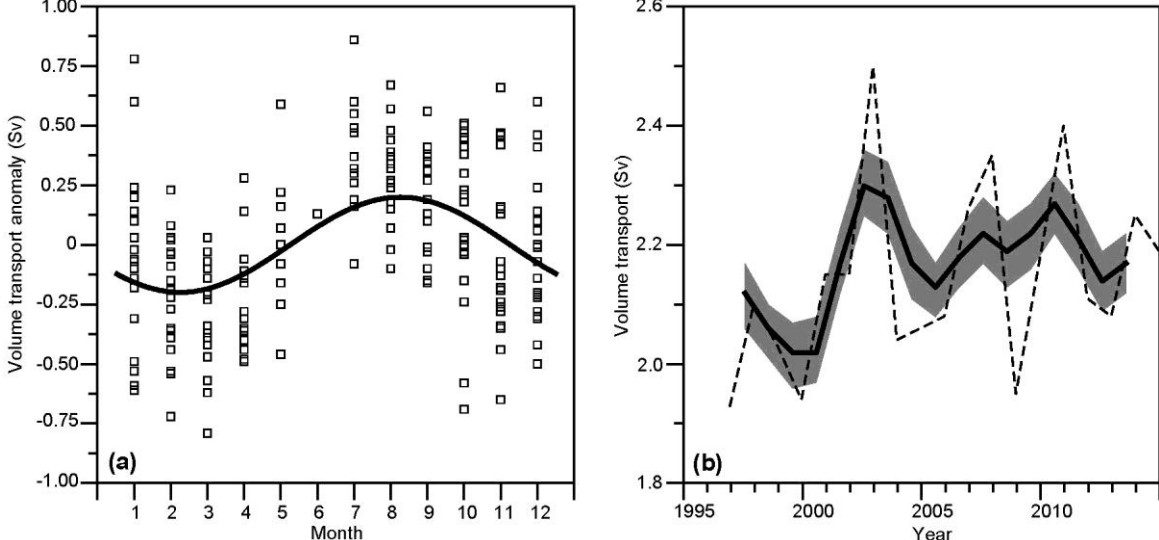

**Figure 3.** Seasonal **(a)** and long-term **(b)** variations of kinematic overflow 1995-2015. **(a)** Each square represents transport deviation for one month from the 3-year running mean. The curve represents the iteratively determined sinusoidal seasonal fit. **(b)** Annually averaged transport excluding days number 136 to 195 (dashed curve) and 3-year running mean transport (continuous curve) with the shaded area representing ± 1 standard error over each 3-year period.




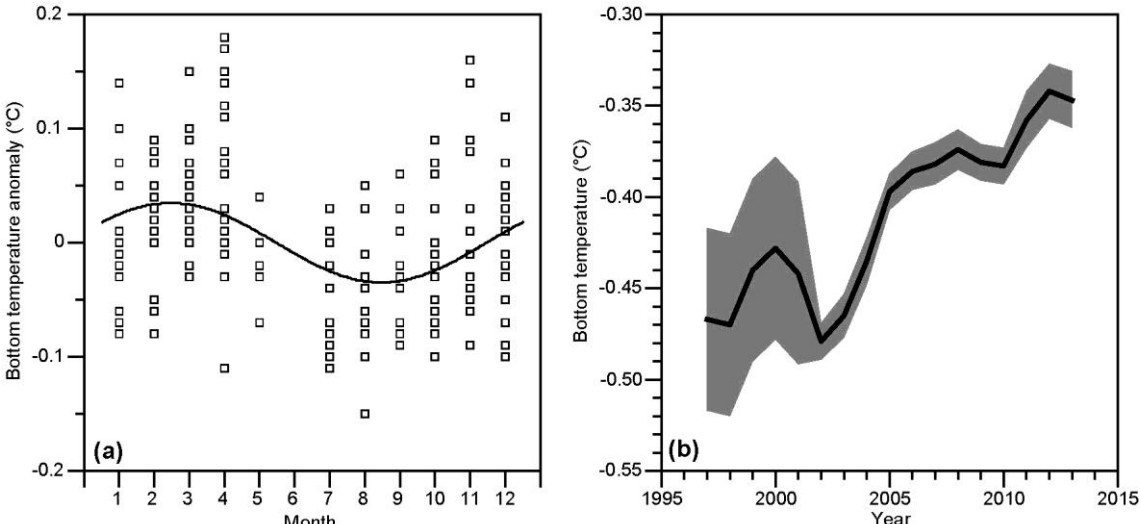

**Figure 4.** Seasonal **(a)** and long-term **(b)** variations of bottom temperature at FB 1995-2015. **(a)** Each square represents the deviation for one month from the 3-year running mean. The curve represents the iteratively determined sinusoidal seasonal fit. **(b)** 3-year averaged transport (black curve) with the shaded area representing the uncertainty interval.





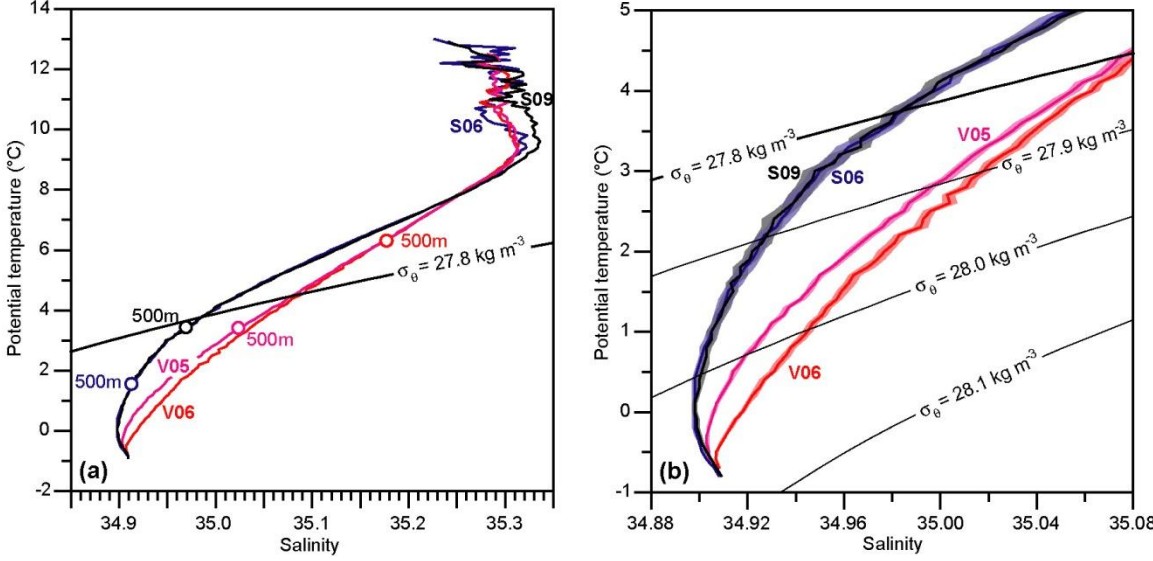

**Figure 5.** θ-S diagrams for four standard stations based on 73 occupations at stations V05 (magenta) and V06 (red) simultaneously and on 54 occupations at stations S06 (blue) and S09 (black) simultaneously in the period 1996 – 2010. **(a)** Average θ-S traces for the whole water column with conditions at 500 m depth indicated by circles. **(b)** Expanded view of waters colder than 5 °C with the average for each station shown as a coloured line surrounded by a shaded area in the same colour representing the average ± 1 standard error. Station locations are shown on Fig. 1 and Fig. 2a.



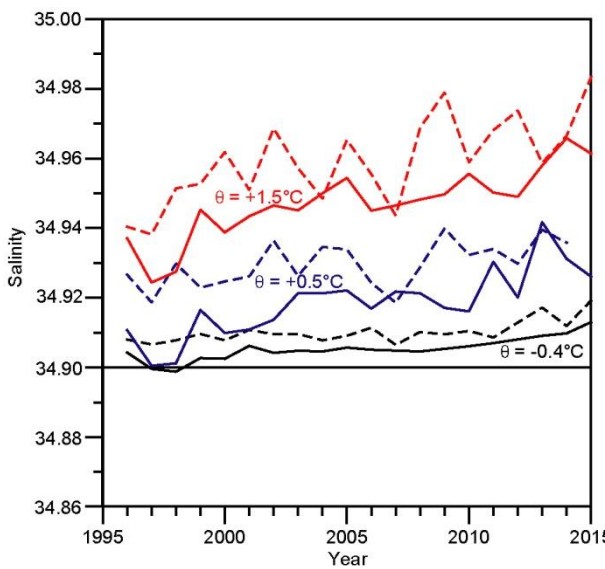

**Figure 6.** Temporal change of salinity at three fixed potential temperatures (θ): -0.4 °C (black), +0.5 °C (blue), +1.5 °C (red) for station V05 (continuous) and V06 (dashed).





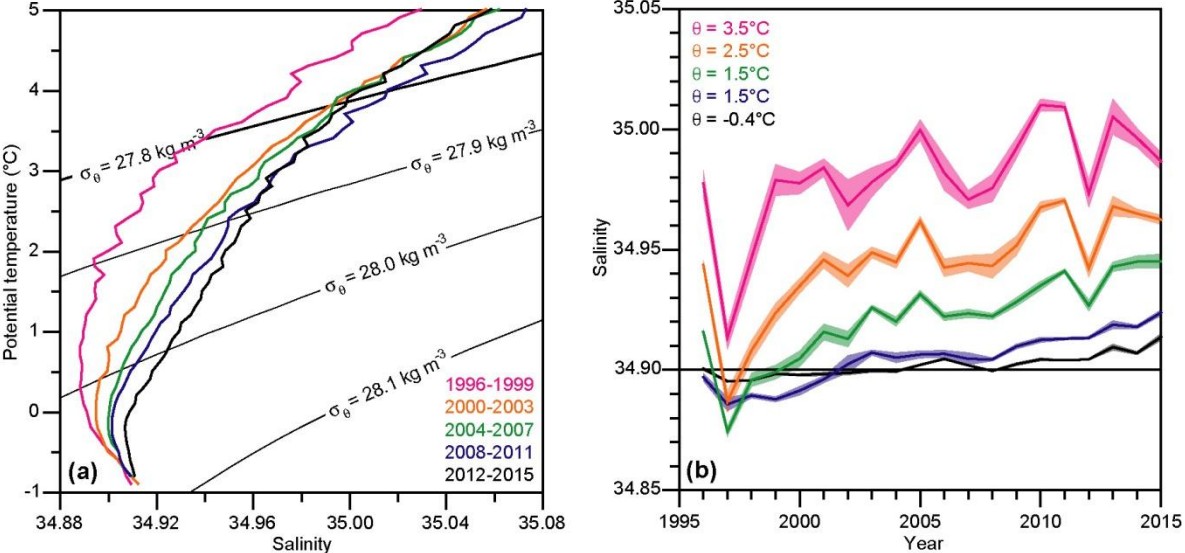

**Figure 7.** Temporal variations of the water mass characteristics in the deep parts of the FSC. **(a)** θ-S traces averaged over consecutive 4-year periods. **(b)** Salinity trends for five different potential temperatures where the annual averages are shown
5  by the coloured lines surrounded by shaded areas representing average ± 1 standard error for each year. The figure is based on 505 CTD profiles from area A in the FSC (Fig. 1).




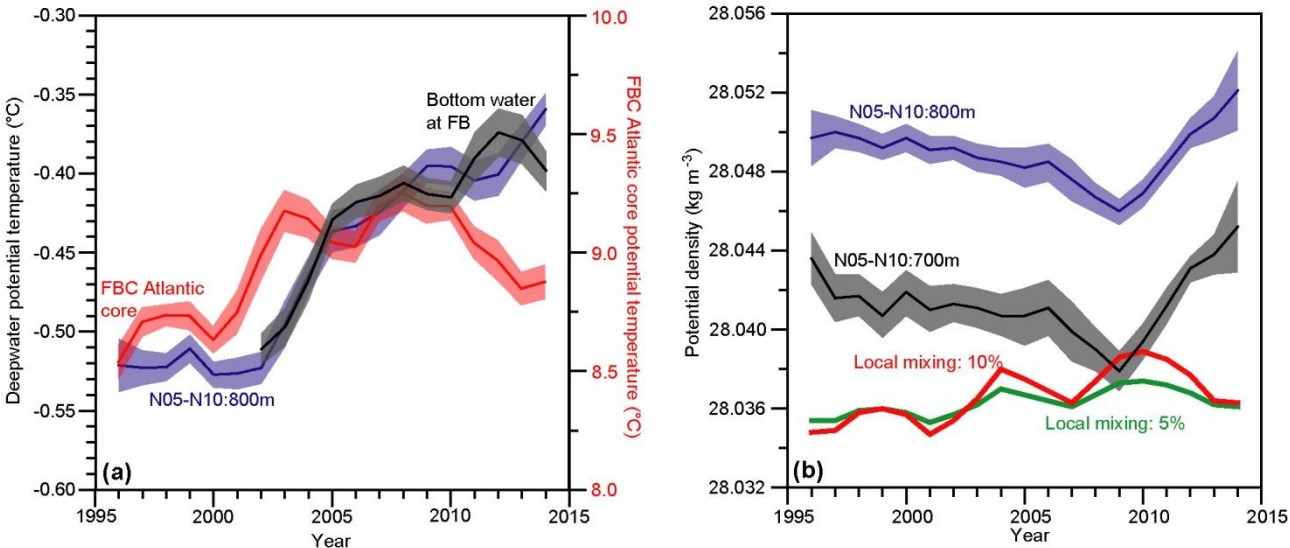

**Figure 8. (a)** Potential temperature of the Atlantic water core on section V (red, right axis), of the bottom water at FB (black, left axis), and at 800 m depth on section N (average of stations N05 to N10, blue, left axis). **(b)** Potential density at two fixed depths (black: 700 m and blue: 800 m) on section N (average of the six stations N05 to N10) and for the overflow layer assuming two different fractions of locally admixed Atlantic core water (green: 5 % and red: 10 %); see text. All curves are 3-year running mean with shaded areas representing average ± 1 standard error over each 3 year period.

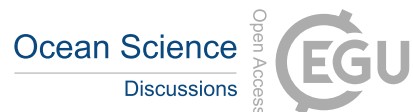

**Figure 9.** Development of overflow plume downstream of the FBC sill. **(a)** Map of the downstream region with gray areas shallower than 500 m. Arrows with temperature indicate average current velocity and temperature 25 m above bottom from 3 July to 5 November 1999 along mooring array A in Geyer et al. (2006) with the arrow over Faroe Bank indicating the velocity scale. Red line indicates section M with circles indicating CTD stations (stations M5, M6, and M7 between M4 and M8 are not shown on map) occupied in May 2016. **(b)** Temperature distribution along section M for the water colder than 3 °C on 20-21 May 2016. **(c)** Vertical temperature variation for the deepest 100 m at the eight CTD stations occupied on section M 20-21 May 2016.