# Peer review of "A stable Faroe Bank Channel overflow 1995-2015"

_Ocean Science, 2016_

## Referee Comment (RC1) · Anonymous Referee #1 · 29 Aug 2016

This manuscript shows, based on in situ observations, that the dense overflow through the Faroe Bank Channel has not increased in the period between 1995 and 2015. Although the scientific contribution of this work is not huge as it is a minor extension of previous work, the paper is extremely well written, well organized, highly relevant and with clear figures, and shows observations that are of interest to the community, and I therefore recommend that this paper should be accepted for publication.

There are really only two small issues, in my opinion, that the authors should clarify or justify. First, in Section 3.3/Figures 6 and 7, the salinity is discussed as a function of potential temperature. I understand that the observations are mainly for temperature, but a far more obvious way to look at this would be to plot potential density (which mainly depends on salinity in this region of the T/S space) against potential temperature. In general, throughout the manuscript with a very few exceptions, the authors attempt to

relate temperature changes to density changes, which just does not make much sense given the nonlinearities in the equation of state. When isotherms and isopycnals are equated (as on p. 13) without any proper justification, the wrong conclusion could be drawn.

Second, the factor of 5-10 on p. 9 confused me. With a 5-10% contribution to the mixture I would expect that changes in the temperature/salinity would be reduced by a factor of 10-20. How is the factor of 5-10 derived?

Textual: - There are quite a few commas in the text where there should be none. - p. 4, l. 1: profiles - p. 4, l. 8: an evaluation cannot draw a conclusion, but it can lead to a conclusion - p. 10, l. 8: close to the bottom

---

## Referee Comment (RC2) · Anonymous Referee #2 · 2 Sep 2016

The manuscript addresses the main question of whether there have been long-term changes in the Faroe Bank Channel (FBC) overflow volume transport or density over the last two decades. In situ measurements from current profiler moorings show that the overflow has been stable in both respects during the period 1995–2015. Because of the role of the overflow as part of the AMOC, and recent attention to the potential weakening of the AMOC, this is an important observation. The short manuscript is well-written and clear, with neat figures, and the topic is interesting and appropriate for this journal. However, I think some improvements could be made to the structure and content of in particular the introduction and discussion sections, before publication.

1. Regarding the structure of the paper: a large part (almost half) of the discussion is dedicated to overflow modification. This is an important topic, and this discussion section (4.3) gives a good overview of literature on the topic. However, the proportion

of the discussion section that is dedicated to overflow modification is surprising considering that this topic is not even alluded to in for example the paper abstract or title. My suggestions would be to a) update the abstract to mention the discussion on overflow modification b) consider shortening this section of the discussion.

2. As mentioned in section 4.3, water mass transformation occurs mainly downstream of the FBC, that is, downstream of the long-term measurements that are the main focus of this manuscript. Some new observations from the downstream region are presented in section 4.3, namely, eight CTD stations occupied 20-21 May 2016. In view of the known high level of short-term variability (oscillations) in this part of the overflow, briefly mentioned in the manuscript on p. 12, L26-29, how does one confidently interpret 8 profiles taken during 2 days? Are the observations in this snapshot representative? Do these measurements add significantly to the substantial body of work done in this region over decades, which includes repeated CTD sections, moorings, etc.? Personally, I am not convinced that the new downstream CTD profiles add enough new information to motivate their inclusion in the manuscript. I recommend focusing on the truly impressive data sets: the time series from the moorings at the sill, and from the repeated standard CTD sections.

3. Speaking about impressive long-term measurements: Is this the first time the whole 20 year time series is presented except in a technical report (Hansen et al., 2015a)? In that case it is a substantial extension of the data set (a doubling of the 10-year time series from e.g. the important HØ2007 paper), and perhaps that should be stated outright in order to make the contribution of this paper clear. If not, other recent manuscript that use all or most of the 2-decade time series ought to be referenced and pointed out.

4. The early years of the FBC overflow time series gave quite a different impression, namely a reduction in the strength of the overflow (Hansen et al., 2004; cited in this manuscript, but only to describe a simple water mass mixing scheme). Even though the earlier conclusion of decreasing overflow [since 1950] (Hansen et al., 2001; not cited in this manuscript) has already been refuted in e.g. Olsen et al., 2008, these

earlier papers and conclusions (as well as papers refuting them) - by these authors and others - are part of the history of the FBC overflow time series, and form an important backdrop to the discussion about the overflow stability. This should be included in the introduction and discussion sections.

Minor comments P2, L 24: study the long-term variations
* * *

---

## Referee Comment (RC3) · Anonymous Referee #3 · 8 Sep 2016

A stable Faroe Bank Channel overflow 1995-2015 by Bogi Hansen, Karin Margretha Husgard Larsen, Hjalmar Hatun, Svein Osterhus

- Summary -

This paper documents the long-term timeseries measurements of the FBC overflow using bottom-mounted ADCP moorings at the sill in conjunction with regular hydrographic sections around the Faroe Islands. Although the moorings do not measure stratification within the water column, the combination of the velocity-determined transport timeseries with the spatial CTD surveys allows the identification of watermass changes and the identification of various possible pathways and processes in this long-term climate record.

The principal conclusion from this analysis is that while no significant change in overflow transport or density has been seen over the 15 years of continuous monitoring, overflow temperature and salinity have both increased, likely due to 5-10 year earlier changes in the Atlantic Water inflow to the Nordic Seas.

- General Comments -

Because of recent observations by the RAPID array of weakening in the Atlantic Meridional Overturning Circulation (AMOC), this is a good time to revisit the long-term FBC record for evidence of similar changes. The paper does a nice job summarizing the FBC monitoring efforts and combining the moored measurements with the hydrographic surveys.

In general, the paper is well thought out and well constructed, with high-quality and adequately described figures (including those in both the primary document and in the supplement–which are really essential for the understanding of the manuscript). In addition, there is a well-written discussion section reviewing recent literature and the current state of knowledge of the role of the FBC in the AMOC. This section considers various possible connections between the FBC overflow and the southward and northward branches of the overturning and is a valuable contribution to the scientific debate.

A few points (including the discussions of mixing and the brief presentation of new cruise results in Fig.9) are somewhat tangential to the main thrust of the paper, but all of these are interesting and relate in some way to the scientific discussion.

My principal criticisms as a scientific reviewer center on the need for additional analysis of the "kinetic overflow" approach (described in HO2007) to better define random and bias errors, as well as the sensitivity to particular parameter choices. With this manuscript's renewed focus on long-term changes (and a longer companion hydrographic dataset) there is an opportunity to develop increased confidence in the data quality and interpretation. 1) HO2007 developed the "kinematic overflow" (KO) approach required by the lack of simultaneous velocity and CTD measurements and established that (a) the velocitydefined interface does co-vary with a temperature-defined interface (isotherm height), although the relationship is not extremely tight, and (b) velocity at adjacent mooring locations is highly correlated, so that a single mooring could be used to represent the flow through the entire channel. However, both of these relationships introduce some error into the final transport (with an unknown level of reduction due to averaging when computing standard errors).

When investigating long-term trends, the KO calculation is most vulnerable to trends in these possible bias errors, so it is important to construct timeseries of (a) the difference between the annually-averaged velocity interface and a particular isotherm height (e.g., 7 degrees), and (b) cross-stream gradients in hydrographic temperature, density, or isotherm slope.

2) The principal temperature timeseries presented is from the near-bottom measurement at the ADCP location, but more relevant for the AMOC would be the average properties (T, S, and density) of the overflow layer. Apparently no attempt has been made to compute these (using, for example, annual averages from the hydrographic sections), although the relationship between bottom temperature and interface height has been presented in HO2007. From the T-S changes presented in Fig.7, it is clear that the overflow layer has undergone changes, but how do these impact the layer average (using interface definitions based on density, temperature, depth, or average velocity profile)?

3) One particular hole in the KO analysis is the missing transport above the selected interface (the height where the velocity drops to 50% of the maximum). HO2007 pointed out that outflowing water above this level could include contributions from both dense overflow and entrained Atlantic Water from above, claiming that the overflow water in the layer is likely compensated by Atlantic Water below the interface. For volume budgets, all outflowing fluid needs to be included, while from the watermass perspective,

СЗ

a density-anomaly-weighted transport might be more appropriate. Differences among these choices could have a large impact on the detection of small trends in temporal variability in the presence of a large annual cycle and monthly wind-forced variability. Therefore, it is important to investigate the long-term trends in all of these neglected components.

4) One issue that has not been mentioned in any of the papers or tech reports by the Torshavn/Bergen group is an intermittent contamination and low-velocity bias in 75KHz ADCP data, apparently caused by side-lobe bottom reflections, that has recently been discovered by the Hamburg group maintaining the Denmark Strait transport moorings. See the Quadfasel, Jochumsen, et al, presentation at the Feb 2015 NACLIM meeting linked here: http://naclim.zmaw.de/fileadmin/user\_upload/naclim/Archive/Meetings/ Annual\_meeting\_2015/PPT/S1.2-1\_Detlef\_Q\_Overview.pdf

Although the issue seems to be most pressing in the Denmark Strait locations, there is a suggestions that the same issue could at least occasionally influence the FBC moorings. Has any attempt been made to quantify and/or eliminate this? The DS issues seem to vary with instrument version (especially internal processing algorithms), mounting hardware configuration, and local bottom properties. The possibility is important enough that should be addressed (if not in this publication, then another upcoming one) even if the FBC dataset does not require the kind of major corrections applied to the DSO measurements.

- Specific Comment -

p.5,I.6. What is meant by the statement that a barotropic current could introduce a bias in the transport? The moored ADCP measures absolute velocity and is not vulnerable to a level-of-no-motion assumption. Is this referring to the fact that the interface (arbitrarily defined as the level at which the current speed is 50% of the max) will be shifted by a barotropic current? (It will, but so will the true transport.) Or is it related to the possibility of barotropic recirculation making the mooring location less representative

of the average transport through the channel. This is indeed a possibility, but can't be diagnosed from measurements at the mooring alone.

- Conclusion -

Since the approach presented here is clearly documented and has been explored from a number of angles, I don't feel that a large amount of revision should be required for publication of the current work. However, the lack of sensitivity analysis on the KO formula and the remaining un-pursued lines of investigation into possible KO biases described above (including water above the velocity-defined interface, velocity-temperature interface differences, and cross-stream gradients) make this unique long-term dataset weaker than it could otherwise be. I'd encourage the authors to follow up these issues.

For example, my calculation "by eye" from Fig.8 of HO2007 suggests that the missing transport above the interface could be 10-15% of the total, and this layer could easily have long-term variability distinct from the lower layer. Certainly, a better estimate than mine can be made from the data.

---

## Author Comment (AC1) · 11 Oct 2016

General response The comments by the three referees have been very constructive and positive. We have tried to address them all as detailed in our responses to individual comments below and as carried out in the changes made in the revised manuscript. In our responses, we refer to page and line numbers in the revised manuscript, where all but the smallest text corrections (except comma deletions or similar) are red.

In addition to the changes suggested by the referees, we have made some minor changes to a few phrases for better readability. In addition, we have added a new paragraph to the beginning of Sect. 4.4, which relates our results to the global energy budget (p. 15, l. 25-32 in revised manuscript). This is an obvious and important implication of our results that we should have mentioned in the original manuscript.

[Figure]

We have also added brief references to this to the abstract (p. 1, l. 16, 20, 26 in revised manuscript) and a reference (p. 19, l. 22-24 in revised manuscript).

Anonymous Referee #1

Comment 1.1: First, in Section 3.3/Figures 6 and 7, the salinity is discussed as a function of potential temperature. I understand that the observations are mainly for temperature, but a far more obvious way to look at this would be to plot potential density (which mainly depends on salinity in this region of the T/S space) against potential temperature. In general, throughout the manuscript with a very few exceptions, the authors attempt to relate temperature changes to density changes, which just does not make much sense given the nonlinearities in the equation of state. When isotherms and isopycnals are equated (as on p. 13) without any proper justification, the wrong conclusion could be drawn.

Response: The referee is correct that for the cold and deep part of the overflow (but not the upper part), density is determined more by salinity than temperature and some of our text seemed to neglect that. In the revised manuscript, this should now be removed. It is not clear to us what the referee suggests by the words: "to plot potential density (which mainly depends on salinity in this region of the T/S space) against potential temperature" since salinity is changing with time. In any case, it appears that our reason for presenting Figures 6 and 7 has not been clear in the original manuscript. We have rewritten the text introducing these figures (p. 8, l. 30 – p. 9, l. 5 in revised manuscript). Hopefully, this addresses the concern raised by the referee.

Comment 1.2: Second, the factor of 5-10 on p. 9 confused me. With a 5-10% contribution to the mixture I would expect that changes in the temperature/salinity would be reduced by a factor of 10-20. How is the factor of 5-10 derived?

Response: The Referee is absolutely correct. We have corrected this embarrassing error. (p. 10, l. 5 and p. 13, l. 13 in revised manuscript).
Comment 1.3: Textual: - There are quite a few commas in the text where there should be none. - p. 4, l. 1: profiles - p. 4, l. 8: an evaluation cannot draw a conclusion, but it can lead to a conclusion - p. 10, l. 8: close to the bottom

Response: We have corrected these

Anonymous Referee #2

Comment 2.1: Regarding the structure of the paper: a large part (almost half) of the discussion is dedicated to overflow modification. This is an important topic, and this discussion section (4.3) gives a good overview of literature on the topic. However, the proportion of the discussion section that is dedicated to overflow modification is surprising considering that this topic is not even alluded to in for example the paper abstract or title. My suggestions would be to a) update the abstract to mention the discussion on overflow modification b) consider shortening this section of the discussion.

Response: Following the recommendation of the referee, we have shortened the discussion on overflow modification by removing reference to the new CTD observations (see response to comment 2.2). In addition, we added a sentence on overflow modification to the abstract (p. 1, l. 19-21 in revised manuscript).

Comment 2.2: As mentioned in section 4.3, water mass transformation occurs mainly downstream of the FBC, that is, downstream of the long-term measurements that are the main focus of this manuscript. Some new observations from the downstream region are presented in section 4.3, namely, eight CTD stations occupied 20-21 May 2016. In view of the known high level of short-term variability (oscillations) in this part of the overflow, briefly mentioned in the manuscript on p. 12, L26-29, how does one confidently interpret 8 profiles taken during 2 days? Are the observations in this snapshot representative? Do these measurements add significantly to the substantial body of work done in this region over decades, which includes repeated CTD sections, moorings, etc.? Personally, I am not convinced that the new downstream CTD profiles add enough new information to motivate their inclusion in the manuscript. I recommend

focusing on the truly impressive data sets: the time series from the moorings at the sill, and from the repeated standard CTD sections.

Response: We have followed this advice and deleted Sect. 3.4 and Fig. 9 and also almost all reference to these observations in Sect. 4.3; only retaining a brief reference to "unpublished data" in lines 22-25 on page 14 of the revised manuscript.

Comment 2.3: Speaking about impressive long-term measurements: Is this the first time the whole 20 year time series is presented except in a technical report (Hansen et al., 2015a)? In that case it is a substantial extension of the data set (a doubling of the 10-year time series from e.g. the important HØ2007 paper), and perhaps that should be stated outright in order to make the contribution of this paper clear. If not, other recent manuscript that use all or most of the 2-decade time series ought to be referenced and pointed out.

Response: This is now done in the revised manuscript (p. 3, l. 9).

Comment 2.4: The early years of the FBC overflow time series gave quite a different impression, namely a reduction in the strength of the overflow (Hansen et al., 2004; cited in this manuscript, but only to describe a simple water mass mixing scheme). Even though the earlier conclusion of decreasing overflow [since 1950] (Hansen et al., 2001; not cited in this manuscript) has already been refuted in e.g. Olsen et al., 2008, these earlier papers and conclusions (as well as papers refuting them) - by these authors and others - are part of the history of the FBC overflow time series, and form an important backdrop to the discussion about the overflow stability. This should be included in the introduction and discussion sections.

Response: We have added a paragraph on this to the introduction (p. 3, l. 5-8 in the revised manuscript) and added the reference (p. 18, l. 18-19 in revised manuscript).

Comment 2.5: Minor comments P2, L 24: study the long-term variations

Response: The word "long-term" was inserted (p. 2, l. 26 in the revised manuscript).

Anonymous Referee #3 Comment 3.1: A few points (including the discussions of mixing and the brief presentation of new cruise results in Fig.9) are somewhat tangential to the main thrust of the paper, but all of these are interesting and relate in some way to the scientific discussion.

Response: We agree that Fig. 9 and some of the associated text was rather tangential and have followed the advice of Referee #2 to delete it, keeping only a brief reference (See our response to Comment 2.2).

Comment 3.2: My principal criticisms as a scientific reviewer center on the need for additional analysis of the "kinetic overflow" approach (described in HO2007) to better define random and bias errors, as well as the sensitivity to particular parameter choices. With this manuscript's renewed focus on long-term changes (and a longer companion hydrographic dataset) there is an opportunity to develop increased confidence in the data quality and interpretation. HO2007 developed the "kinematic overflow" (KO) approach required by the lack of simultaneous velocity and CTD measurements and established that (a) the velocitydefined interface does co-vary with a temperature-defined interface (isotherm height), although the relationship is not extremely tight, and (b) velocity at adjacent mooring locations is highly correlated, so that a single mooring could be used to represent the flow through the entire channel. However, both of these relationships introduce some error into the final transport (with an unknown level of reduction due to averaging when computing standard errors). When investigating long-term trends, the KO calculation is most vulnerable to trends in these possible bias errors, so it is important to construct timeseries of (a) the difference between the annually-averaged velocity interface and a particular isotherm height (e.g., 7 degrees), and (b) cross-stream gradients in hydrographic temperature, density, or isotherm slope.

Response: Referee #3 raises some very relevant issues and we have tried to address them by including a new table (Table 3) and additional text at the end of Sect. 4.1 (p. 11, l. 8 – p. 12, l. 6 in the revised manuscript) and also two tables (Tables S1 and S2) and a figure (Fig. S9) in the supplementary document. We feel that this has

strengthened our conclusions and have replaced "kinematic overflow" with the more general "overflow volume transport" in the heading of Sect. 4.1 and elsewhere (p. 10, l. 26, p. 11, l. 1, p.12, l. 8 in the revised manuscript)

Comment 3.3: The principal temperature timeseries presented is from the near-bottom measurement at the ADCP location, but more relevant for the AMOC would be the average properties (T, S, and density) of the overflow layer. Apparently no attempt has been made to compute these (using, for example, annual averages from the hydrographic sections), although the relationship between bottom temperature and interface height has been presented in HO2007. From the T-S changes presented in Fig.7, it is clear that the overflow layer has undergone changes, but how do these impact the layer average (using interface definitions based on density, temperature, depth, or average velocity profile)?

Response: Again, the referee has made a very useful recommendation that we have addressed and which we feel strengthens the manuscript substantially. To address this issue, we have added Table 4 and associated text into Sect. 4.2 (p. 13, l. 3-8 in the revised manuscript).

Comment 3.4: One particular hole in the KO analysis is the missing transport above the selected interface (the height where the velocity drops to 50% of the maximum). HO2007 pointed out that outflowing water above this level could include contributions from both dense overflow and entrained Atlantic Water from above, claiming that the overflow water in the layer is likely compensated by Atlantic Water below the interface. For volume budgets, all outflowing fluid needs to be included, while from the watermass perspective, a density-anomaly-weighted transport might be more appropriate. Differences among these choices could have a large impact on the detection of small trends in temporal variability in the presence of a large annual cycle and monthly wind-forced variability. Therefore, it is important to investigate the long-term trends in all of these neglected components.

Response: In this manuscript we have not attempted to treat the FBC-overflow in relation to volume budgets and we believe that question to require considerably more extensive observations and analysis. The referee is, however, justified in questioning whether variations in the layer above our defined interface could influence our conclusions. In the revised manuscript we have tried to answer this by considering the effect of two alternative definitions of kinematic overflow (p.5, l. 17-27 and p. 11, l. 5-7 in revised manuscript + supplement Fig. S2). Although average transport values may be affected by this, we find that the long-term variations and trend are not.

Comment 3.5: One issue that has not been mentioned in any of the papers or tech reports by the Torshavn/Bergen group is an intermittent contamination and low-velocity bias in 75KHz ADCP data, apparently caused by side-lobe bottom reflections, that has recently been discovered by the Hamburg group maintaining the Denmark Strait transport moorings. See the Quadfasel, Jochumsen, et al, presentation at the Feb 2015 NACLIM meeting linked here: http://naclim.zmaw.de/fileadmin/user_upload/naclim/Archive/Meetings/ Annual_meeting_2015/PPT/S1.2-1_Detlef_Q_Overview.pdf. Although the issue seems to be most pressing in the Denmark Strait locations, there is a suggestions that the same issue could at least occasionally influence the FBC moorings. Has any attempt been made to quantify and/or eliminate this? The DS issues seem to vary with instrument version (especially internal processing algorithms), mounting hardware configuration, and local bottom properties. The possibility is important enough that should be addressed (if not in this publication, then another upcoming one) even if the FBC dataset does not require the kind of major corrections applied to the DSO measurements.

Response: We are aware of this problem, but do not believe that it affects our results. To our knowledge, this problem arises mainly for some RDI Long Ranger ADCPs. We have mainly used RDI Broadband ADCPs and do not see similar symptoms. The only exceptions are a Long Ranger deployment at FB from September 2012 to May 2013 and a Long Ranger deployment at FG from May 2008 to May 2009. The first of these deployments did indeed show similar behavior to that seen by the Hamburg group in the Denmark Strait, as documented in a technical report: http://www.hav.fo/PDF/Ritgerdir/2014/TecRep1401.pdf. Fortunately, there was a Broadband ADCP at the same site, which originally was assumed to be lost, but was later recovered. Thus, we have not used the data from the Long Ranger at FB. In the other case, the Long Ranger at FG was in a trawl-proof frame, which keeps instrument tilt very small and may also block side-lobes. Whether that or different firmware is the explanation, this system has not shown these symptoms neither during this deployment nor during other deployments in overflow regions (e.g., Olsen et al., 2016).

Comment 3.6: p.5,l.6. What is meant by the statement that a barotropic current could introduce a bias in the transport? The moored ADCP measures absolute velocity and is not vulnerable to a level-of-no-motion assumption. Is this referring to the fact that the interface (arbitrarily defined as the level at which the current speed is 50% of the max) will be shifted by a barotropic current? (It will, but so will the true transport.) Or is it related to the possibility of barotropic recirculation making the mooring location less representative of the average transport through the channel. This is indeed a possibility, but can't be diagnosed from measurements at the mooring alone.

Response: This paragraph was not well phrased and has been deleted from the revised manuscript. The problem is now treated in a different way in the last two paragraphs of Sect. 2.3 (p. 5, l. 17-27 in revised manuscript and Fig. S2 in revised supplement).

Comment 3.7: Since the approach presented here is clearly documented and has been explored from a number of angles, I don't feel that a large amount of revision should be required for publication of the current work. However, the lack of sensitivity analysis on the KO formula and the remaining un-pursued lines of investigation into possible KO biases described above (including water above the velocity-defined interface, velocitytemperature interface differences, and cross-stream gradients) make this unique longterm dataset weaker than it could otherwise be. I'd encourage the authors to follow

up these issues. For example, my calculation "by eye" from Fig.8 of HO2007 suggests that the missing transport above the interface could be 10-15% of the total, and this layer could easily have long-term variability distinct from the lower layer. Certainly, a better estimate than mine can be made from the data.

Response: We agree that the original manuscript was lacking in this regard and thank the referee for the detailed comments. We hope that the revised manuscript (see responses to comments above) has clarified these issues.

Please also note the supplement to this comment:
http://www.ocean-sci-discuss.net/os-2016-56/os-2016-56-AC1-supplement.pdf

**Supplement:**

[Figure]

**Figure S1.** A typical velocity (towards 304°) profile at site FB. The interface at each instant is defined to be at the height where the velocity has decreased to 50 % of the core velocity ($V_{max}$).

[Figure]

**Figure S2.** Annually averaged (excluding the period from day 136 to day 195) transport density at ADCP site FB, defined as the vertically integrated velocity from the deepest measurement (bin 1) up to the interface, with the interface defined in three different ways: the *standard* interface (black curve), the *baroclinic* interface (red curve), and the *fixed* interface (blue curve). See the last paragraph of Sect. 2.3 in the main manuscript for definitions of these three interfaces.

[Figure]

**Figure S3.** Monthly averaged kinematic overflow through the FBC for months with at least 28 days of observations at site FB.

[Figure]

**Figure S4.** **(a)** Vertical temperature variation at CTD station V06 close to the sill depth. The black line shows the average difference between temperature at a given depth and simultaneous temperature at 840 m depth. Shaded area indicates ± 1 standard error. Based on 68 CTD profiles at V06. **(b)** The bottom temperature at FB plotted against simultaneous temperature at 810 m depth at V06. Open squares are from the period before the use of Microcats at FB in 2001. The diagonal line indicates equality.

[Figure]

**Figure S5.** Temporal variations of potential temperature (red), salinity (blue), and potential density (black) at 800 m depth at station S08, which is the deepest station on section S. Each parameter is shown by a curve following the 3-year running mean surrounded by a shaded area in the same colour representing ± 1 standard error.

[Figure]

**Figure S6. (a)** Potential temperature close to the bottom at site FB (black, left axis), at 800 m on section N (blue, left axis), and of the Atlantic water core on section N (red, right axis). **(b)** Salinity at 800 m on section N (blue, left axis), at 800 m at S08 (green, left axis), and of the Atlantic core on section N (red, right axis). Each parameter is shown by a curve following the 3-year running mean surrounded by a shaded area in the same colour representing ± 1 standard error (over 3 years). For the potential bottom temperature at FB, the shaded area includes the instrumental uncertainty. Potential temperature and salinity at 800 m on section N are calculated as the average of six stations (N05 to N10) on 91 cruises from 1991 to 2015 (blue).

[Figure]

**Figure S7.** Potential temperature (red) and salinity (blue) of the Atlantic water core on section V. Each parameter is shown by a curve following the 3-year running mean surrounded by a shaded area in the same colour representing ± 1 standard error.

[Figure]

**Figure S8.** Comparison of long time series for Atlantic water properties in the FSC and deep water properties at station M. Temperature **(a)** and salinity **(b)** of NAW and MNAW in the FSC and at two depths at station M. Data from the FSC have been downloaded from www.ices.dk. Data from station M are available at the Norwegian Marine Data Centre (www.nmdc.no).

[Figure]

**Figure S9.** Depth of the $\sigma_\theta = 27.8$ kg m$^{-3}$ isopycnal at station V06 plotted against the depth of the same isopycnal at V05 from 85 cruises where the two stations have been occupied within a day from one another. The regression line shown had a correlation coefficient of 0.78.

**Table S1.** Correlation between the depth of the interface at FB and bottom temperature at FG from June 2008 to May 2009 after averaging over three different periods.

| Averaging period: | 1 day | 7 days | 31 days |
|---|---|---|---|
| Number of values: | 343 | 49 | 11 |
| Correlation coeff.: | 0.42 | 0.43 | 0.64 |

**Table S2.** Correlation coefficient (R) between the depth of the interface at FB and the depth of the $\sigma_\theta = 27.8$ kg m$^{-3}$ isopycnal at V05 or V06 either on the same day (Lag = 0) or lagged by one day. N is the number of values.

| CTD station | Lag | N | R |
|---|---|---|---|
| V05 | 0 | 73 | 0.42 |
| V05 | 1 day | 75 | 0.43 |
| V06 | 0 | 70 | 0.57 |
| V06 | 1 day | 73 | 0.73 |